# Cost effectiveness of rituximab and mycophenolate mofetil for neuromyelitis optica spectrum disorder in Thailand: Economic evaluation and budget impact analysis

**Saharat Aungsumart** \*, Metha Apiwattanakul

Neuroimmunology Unit, Department of Neurology, Prasat Neurological Institute, Bangkok, Thailand

* Saharatau@hotmail.com

## Abstract

Neuromyelitis optica spectrum disorder (NMOSD) is an inflammatory condition of the central nervous system. The extent of disability depends on the severity of the disease and the number of relapses. Although azathioprine is currently the main treatment for patients with NMOSD in Thailand, patients often relapse during its use. Hence, it is argued that there are other drugs that would be more effective. The purpose of this study is to evaluate, from a societal perspective and from the economic impact on Thailand's healthcare system, the cost utility of treatment with mycophenolate mofetil (MMF) and rituximab in patients resistant to azathioprine. The Markov model with a one-year cycle length was applied to predict the health and cost outcomes in patients with NMOSD over a lifetime. The results showed that rituximab exhibited the highest quality-adjusted life year (QALY) gains among all the options. Among the rituximab-based treatments, the administration of a rituximab biosimilar with $CD27^+$ memory B cell monitoring proved to be the most cost-effective option. At the willingness-to-pay threshold of 160,000 Thai baht (THB), or 5,289 US dollar (USD), per QALY gained, the treatment exhibited the highest probability of being cost effective (48%). A sensitivity analysis based on the adjusted price of a generic MMF determined that the treatment was cost effective, exhibiting an incremental cost-effectiveness ratio of -164,653 THB (-5,443 USD) and a 32% probability of being cost effective. The calculated budget impact of treating patients resistant to conventional therapy was 1–6 million THB (33,000–198,000 USD) for the first three years, while after the third year, the budget impact stabilized at 3–4 million THB (99,000–132,000 USD). These data indicate that, in Thailand, treatment of drug resistant NMOSD with a rituximab biosimilar with $CD27^+$ memory B cell monitoring or treatment with a generic MMF would be cost effective and would result in a low budget impact. Therefore, the inclusion of both the rituximab biosimilar and a generic MMF in the National Drug List of Essential Medicine for the treatment of NMOSD may be appropriate.

**Data Availability Statement:** All relevant data are within the manuscript and its Supporting Information files.

**Funding:** This project received financial support from the Prasat Neurological Institute, Department of Medical Services, Ministry of Public Health Thailand. The funder had no role in study design, data collection and analysis, decision to publish, or preparation of the manuscript.

**Competing interests:** The authors have declared that no competing interests exist.

## Introduction

Neuromyelitis optica spectrum disorder (NMOSD) is a devastating central nervous system (CNS) inflammatory demyelinating disease that is caused by autoantibodies targeting aquaporin-4 immunoglobulin G (AQP4-IgG) [1]. Patients usually present with severe optic neuritis and myelitis, which can cause blindness and quadriplegia [2]. The extent of the disability depends on the number and severity of relapses. Therefore, the mainstay of therapy is effective relapse prevention and aggressive treatment during attacks. Furthermore, severe attacks are typically managed by treatment with high dose steroids followed by a plasma exchange to rescue neurological function [3]. Accordingly, the cost of treatment is higher in patients with acute severe attacks compared to those with mild attacks for whom high dose steroid therapy is usually sufficient. Moreover, the efficacy of plasma exchange is limited, as only some patients exhibit fully restored neurological function [3–5]. Thus, relapse prevention with immunosuppressive drugs is the most effective treatment. Commonly used drugs for the prevention of NMOSD relapse include prednisolone, azathioprine, mycophenolate mofetil (MMF), and rituximab [6]. There is evidence that rituximab and MMF exhibit greater efficacy compared to azathioprine [7, 8]. Highly efficacious medications not only reduce the number of relapses but also limit the severity of the relapses [9]. However, due to the high cost of rituximab and MMF, azathioprine is the only drug included on the National Drug List of Essential Medicine (NLEM) for the prevention of NMOSD relapses in Thailand. The main objective of this study was to evaluate the cost effectiveness of rituximab and MMF in the treatment of NMOSD patients. The second objective was to estimate the budget required for alternative treatments for NMOSD patients in Thailand.

## Materials and methods

This study used a Markov model to compare the lifetime costs and outcomes of patients with NMOSD undergoing different treatments. Specifically, rituximab and MMF were evaluated in comparison to azathioprine. The study was conducted from a societal perspective, as recommended by the guidelines of the health technology assessment (HTA) of Thailand [10]. The target population consisted of NMOSD patients older than 18 years. In the economic analysis, azathioprine was used as the reference (option 1). In addition, rituximab treatments were classified into two categories, depending on the method used for the administration of the drug. The first regimen (option 2) consisted of a fixed dose of rituximab, which began with the introduction of two 1000 mg intravenous rituximab doses two weeks apart followed by 1000 mg intravenous rituximab doses every six months. The other method, which was based on the monitoring of $CD27^+$ memory B cells (option 3), consisted of the administration of 375 mg/$m^2$ rituximab every week for four weeks, along with the monitoring of $CD27^+$ memory B cells every three months. If the $CD27^+$ memory B cell counts in the peripheral blood exceeded 0.05% of the total mononuclear cells, the patient was infused with additional rituximab at 375 mg/$m^2$ [11]. Moreover, since an equally effective rituximab biosimilar is available in Thailand at a lower price, the latter compound was also evaluated using the fixed-dose regimen (option 4) and the method based on dose adjustment following the $CD27^+$ memory B cell assessment (option 5). Finally, the administration of a 2000 mg dose of MMF per day (option 6) was included in the cost-effectiveness analysis, in which the future outcomes and costs were discounted at a rate of 3% to conform to present day values. The data were collected and analyzed with Excel (Microsoft, Redmond, WA, USA). The outcomes were estimated based on quality-adjusted life years (QALYs) gained, and the incremental cost-effectiveness ratio (ICER) was expressed in Thai baht (THB) per QALY gained. The cost-effectiveness was determined by applying a ceiling threshold of 160,000 THB (5,289 USD) per QALY gained, as recommend by

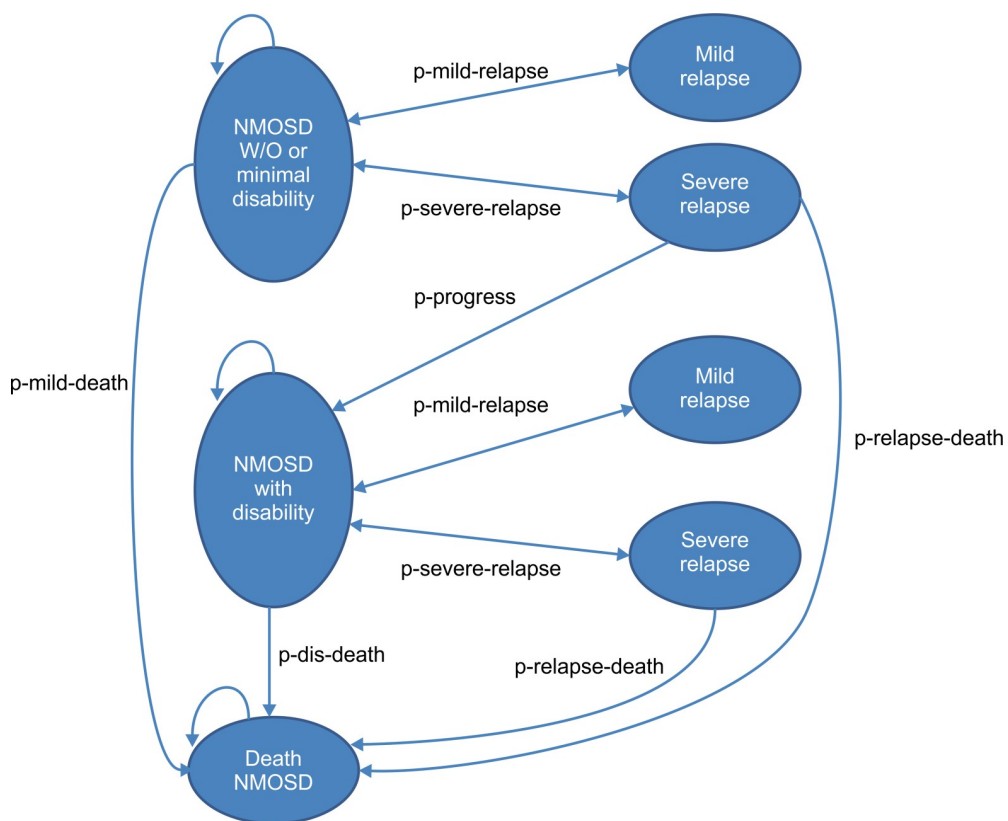

**Fig 1. Schematic Markov model for neuromyelitis optica spectrum disorder (NMOSD) treatment.**
NMOSD = neuromyelitis optica spectrum disorder, W/O = without, p-mild-relapse = probability of current health
stage transitioning to mild relapse, p-severe-relapse = probability of current health stage transitioning to severe relapse,
p-mild-death = probability of patients without or minimal disability stage transitioning to death, p-dis-
death = probability of patients with disability stage transitioning to death, p-relapse-death = probability of patient
death due to severe relapse, and p-progress = probability of patients with no or minimal disability stage transitioning
to moderate or severe disability stage after severe relapse.

the Subcommittee for the Development of the National List of Essential Medicines (NLEM)
[10].

## Economic model

The Markov model shown in Fig 1 was used to simulate the clinical course of patients with
NMOSD and to estimate the health outcomes and costs over a lifetime horizon based on a
cycle length of one year. The study compared five different treatment options, with azathio-
prine as the reference. Three main health states based on the EDSS were considered, namely,
(i) patients with no or mild disability (Expanded Disability Status Scale (EDSS) 0–5.5); (ii)
patients with moderate to severe disability (EDSS 6–9.5); and (iii) deceased NMOSD patients.
In addition, two health stages that were represented by NMOSD patients temporarily
experiencing a mild or severe relapse (temporary stage) were included in the model. These
conditions reflect the natural history of patients with NMOSD. Furthermore, relapse classifica-
tion according to severity was justified by both clinical and economic considerations. Patients
exhibiting a severe relapse were defined by a severe disability that was sustained or worsened
after taking high dose steroids as indicated by EDSS scores $\geq$ 7.0 in patients who presented
with myelitis or in patients with a visual acuity worse than 20/200 who presented with optic

neuritis (4). These groups require more expensive treatments and exhibit lower utility compared to those diagnosed with mild relapse. The temporary stage group included (iv) patients with no or mild disability and mild relapse, (v) patients with no or mild disability and severe relapse, (vi) patients with moderate or severe disability and mild relapse, and (vii) patients with moderate or severe disability and severe relapse. The arrows in Fig 1 denote permissible transitions.

There are several assumptions inherent to this model. First, relapse is classified according to severity. A patient undergoing a severe relapse requires treatment with high dose steroids plus a plasma exchange and frequent physiotherapy, thus implying higher costs compared to patients for whom high dose steroids alone are sufficient. Second, the same probability of relapse was assumed for all the patient groups. Third, whereas NMOSD patients experiencing mild relapse return to the previous health stage after treatment, some patients undergoing severe relapse die while others progress to severe disability. Fourth, surviving patients with moderate or severe disability who experience severe relapse return to the previous health stage. Fifth, patients with moderate or severe disability cannot return to the state of no or mild disability, thus indicating a confirmation of disability progression following a severe relapse.

## Cost variables

Data regarding the direct medical and nonmedical costs of patients with NMOSD were collected at the Prasat Neurological Institute, a tertiary neurological referral center [12]. In brief, this was a cross-sectional study for which patients were recruited between November 1, 2015, and June 30, 2016. Of the 36 patients included in this study, 87% were AQP4 antibody-positive. The average age and average age at onset were 48.48 ± 12.00 and 39.48 ± 12.24 years, respectively. The percentage of female patients was 95.4%. The annualized relapse rate (ARR) was 0.53 ± 0.29. Twenty-five patients diagnosed with severe relapse NMOSD required a plasma exchange, four patients exhibited a mild relapse, and seven patients exhibited no relapse. Because 13% of the cohort were seronegative AQP4 patients, there were some differences in the demographic data of these patients compared with the entire cohort. These differences include a higher percentage of male patients (50%), lower average age and lower average age at onset, i.e., 30.75 ± 8.90 and 24.75 ± 7.9 years, respectively, and lower average ARR (0.35 ± 0.17) compared to the entire cohort. However, the severity of relapse of the seronegative group as indicated by the EDSS and visual acuity scores at the time of relapse was similar to that of the seropositive groups. This result occurred because the selection process required that the health status of the patients be severe enough to perform plasmapheresis. The consumer price index was used to adjust all the costs to the year 2019 values. The exchange rate of 30.3 THB to one US dollar (USD) was used in this study.

Drug costs were obtained from the reference price database of Thailand [13], while other direct medical costs were retrospectively retrieved from the electronic medical records of the patients. All charges were adjusted to a cost-to-charge ratio of 1.63[10]. The costs of hospitalization due to relapse were classified according to the severity of the attack. The direct nonmedical costs, such as food, accommodations, transportation, and formal and informal care, were collected through interviews with patients via a structured questionnaire. To avoid double counting and based on the Thai HTA guidelines, the indirect costs were not included in this study [10].

## Clinical variables

The efficacy of azathioprine relative to rituximab in preventing relapse and the associated probability of achieving a relapse-free status were obtained from a single randomized control

study published by Nikoo et al. [14]. Although a systematic review and meta-analysis regarding the ability of rituximab to prevent NMOSD relapse was published [15], we did not use this meta-analysis due to the heterogeneity of the various study designs. There was also no information regarding relapse-free cases, and some of the analyses in the aforementioned study were not comparable. Data on the efficacy of MMF compared to azathioprine with respect to the ability to yield a relapse-free status were obtained by a systematic review and meta-analysis, in which five studies were selected and included in the pooled analysis via the Stata v16.0 (StataCorp LLC, College Station, TX, USA) (S1 Fig). The details and flow chart of our study selection are reported in the supplementary S1 Text. Moreover, data on the efficacy of rituximab and MMF in preventing severe relapse, relative to azathioprine, were obtained from a single retrospective study conducted by Jeong I. H. et al. [9].

The probabilities of severe relapse, mild relapse, and death due transitioning to relapse were obtained from the Prasat Neurological Institute between October 1, 2017, and September 30, 2018 [16]. In summary, there were 49 acute NMOSD attacks that consisted of 25 cases of first attack and 24 cases of relapse. Severe relapse occurred in 4 of the 24 cases, and one patient died due to a severe attack. All of these NMOSD patients were AQP4 antibody positive. The average age and age at onset were 46.30 ± 11.93 and 42.02 ± 13.43 years, respectively. The percentage of female patients was 93.9%. The ARR was 0.70± 0.41 per year. We posit that the reason to use this cohort rather than the 36 patients in the previous study [12] to determine the probability of transition is because these data were collected in a prospective manner and include all of the NMOSD relapse patients who were treated at the Prasat Neurological Institute during the referenced period. Thus, these data are more realistic for the incidence of relapse, proportion of severe relapse, and incidence of death due to NMOSD relapse. Data on the efficacy of plasmapheresis in Thailand were used to derive the probability of progressing to moderate and severe disability after a severe relapse [4]. The all-cause mortality data in the Thai population by age group were obtained from the life table data provided by the World Health Organization [17]. The standard mortality ratio (SMR) among patients with moderate to severe disability was 2.78 times higher than the SMR in the general population [18].

## Health outcomes

The health outcomes were expressed as QALYs gained and were defined as the number of years spent in each health state multiplied by the utility score. The utility of patients with NMOSD in Thailand was obtained from a previously published multicenter cross-sectional study [19], while that of patients with NMOSD relapse was obtained from another study [12]. In brief, the utility data from 29 relapsing NMOSD patients was obtained through interviews using a Thai version of the EuroQol Five Dimension Questionnaire that included five levels (EQ-5D-5L). The quality of life (QoL) losses due to severe and mild relapse were –0.29 and –0.07, respectively. All the input parameters of the models are shown in Table 1.

## Uncertainty analysis

Probabilistic sensitivity analyses (PSA) were performed to simultaneously test the uncertainty of all the parameters. The Monte Carlo method was run for 1,000 simulations. The results of the PSA were expressed as cost-effectiveness acceptability curves. To identify the best candidate treatment option with respect to cost effectiveness, one-way sensitivity analyses were conducted to evaluate the uncertainty of each parameter. The results are presented as Tornado diagrams.

**Table 1. Cost and input parameters of decision models.**

| Cost and input parameter | Distribution | Mean | Standard deviation | Reference |
|---|---|---|---|---|
| Discounting | | | | |
| Discount rate for costs (%) | | 3 | (0–6) | |
| Discount rate for outcomes (%) | | 3 | (0–6) | |
| Efficacy of medication in relapse prevention | | | | |
| Azathioprine | Beta | 0.41 | 0.242 | [14] |
| MMF | Beta | 0.46 | 0.248 | [7–9,20,21] |
| Rituximab | Beta | 0.65 | 0.228 | [14] |
| Efficacy of medication in prevention of severe relapse | | | | |
| Azathioprine | Beta | 0.17 | 0.139 | [16] |
| MMF | Beta | 0.028 | 0.032 | [9] |
| Rituximab | Beta | 0.014 | 0.017 | [9] |
| Transition probability after severe relapse | | | | |
| Progression to disability after relapse | Beta | 0.190 | 0.02 | [4] |
| Death due to relapse | Beta | 0.02 | 0.002 | [16] |
| Total direct medical cost of treatment in one year (THB) | | | | |
| Severe relapse | Gamma | 395,351 | 111,506 | [12] |
| Mild relapse | Gamma | 34,293 | 6,885 | [12] |
| No relapse | Gamma | 11,401 | 11,332 | [12] |
| Total direct nonmedical cost of treatment in one year (THB) | | | | |
| Severe relapse | Gamma | 24,640 | 4,928 | [12] |
| Mild relapse | Gamma | 7,261 | 1,452 | [12] |
| No relapse | Gamma | 2,650 | 530 | [12] |
| Utility of patients with NMOSD | | | | |
| Normal-mild disability | Beta | 0.515 | 0.010 | [19] |
| Moderate-severe disability | Beta | 0.073 | 0.014 | [19] |
| Disutility after relapse | | | | |
| Mild relapse | Beta | 0.07 | 0.04 | [12] |
| Severe relapse | Beta | 0.29 | 0.07 | [12] |
| Cost of medication in one year (THB) | | | | |
| Azathioprine | Gamma | 3,978 | 795 | [13] |
| MMF | Gamma | 64,240 | 12,848 | [13] |
| Rituximab* (CD27+ memory B cell count monitoring) | Gamma | 194,607[a] | 38,921 | [13] |
| | | 68,869[b] | 19,345 | |
| Rituximab (fixed dose) | Gamma | 145,092[a] | 29,018 | [13] |
| | | 96,728[b] | 19,346 | |
| Rituximab biosimilar* (CD27+ memory B cell count monitoring) | Gamma | 158,070[a] | 31,614 | [13] |
| | | 56,690[b] | 11,338 | |
| Biosimilar Rituximab (fixed dose) | Gamma | 116,070[a] | 23,214 | |
| | | 77,380[b] | 15,476 | [13] |

* Based on CD27+ memory B cell monitoring four times per year with two additional rituximab administrations

[a] Cost of medication in the first year

[b] Cost of medication in the following years

## Budget impact analysis

The budget impact analysis was based on an estimated population of 69 million [22] and on the prevalence rate of NMOSD [23]. The following criteria were used for the analysis: (i)

**Table 2. Costs and outcomes for each medication are expressed in THB (USD).**

| Medication | Total cost THB (USD) | Total effectiveness | | Incremental cost THB (USD) | Incremental effectiveness | | ICER THB (USD) | |
|---|---|---|---|---|---|---|---|---|
| | | LYs | QALYs | | LYs | QALYs | LYs | QALYs |
| Azathioprine | 3,665,371 | 24.29 | 8.40 | | | | | |
| | (120,969) | | | | | | | |
| Rituximab fixed dose | 4,156,051 | 25.49 | 12.31 | 490,681 | 1.20 | 3.91 | 408,455 | 125,530 |
| | (137,163) | | | (16,194) | | | (13,480) | (4,143) |
| Rituximab CD27$^+$ memory B cell regimen | | | | | | | | |
| | 3,453,054 | 25.49 | 12.31 | Dominant | 1.20 | 3.91 | Dominant | Dominant |
| | (113,962) | | | | | | | |
| MMF (2000 mg/day) | 3,746,932 | 25.34 | 11.52 | 81,561 | 1.05 | 3.12 | 77,741 | 26,159 |
| | (123,661) | | | (2,692) | | | (2,566) | (863) |
| Biosimilar of rituximab fixed dose | | | | | | | | |
| | 3,597,221 | 25.49 | 12.31 | Dominant | 1.20 | 3.91 | Dominant | Dominant |
| | (118,720) | | | | | | | |
| Biosimilar of rituximab CD27$^+$ memory B cell regimen | | | | | | | | |
| | 3,097,842 | 25.49 | 12.31 | Dominant | 1.20 | 3.91 | Dominant | Dominant |
| | (102,239) | | | | | | | |

patients resistant to the first-line treatment (azathioprine) who were previously studied at the Prasat Neurological Institute, Thailand [24]; (ii) a treatment coverage rate of 30% for each year; (iii) a treatment coverage rate of 100% in three years (iv); a closed cohort model, and (v) a no cost discount.

## Institution review board

This study was approved by the Institutional Review Board of the Prasat Neurological Institute approval number 61–437207(14). Written informed consent was obtained from all the enrolled patients.

## Results

### Cost-utility analysis

Our model simulated the lifetime of NMOSD patients undergoing six different treatment options. The cumulative costs, QALYs, and ICER are shown in Table 2 and Fig 2. Rituximab had the highest QALY gains compared to the other options. Among the rituximab treatments, the administration of rituximab biosimilar with CD27$^+$ memory B cell monitoring resulted in the lowest lifetime cost, i.e., 3,097,842 THB (102,408 USD). However, these data were based on the assumption that the biosimilar and its administration strategy exhibited similar efficacy to that of the original rituximab. The life years (LYs) gained from the rituximab-based regimen was 25.49, whereas the LYs associated with azathioprine and MMF were 24.29 and 25.34, respectively.

In the cost-effectiveness plane (Fig 2), azathioprine served as the comparator treatment at a fixed point (0, 0). Compared to azathioprine, all the other options except for the fixed-dose original rituximab and MMF, resulted in lower costs and greater QALYs. The administration of the rituximab biosimilar with CD27$^+$ memory B cell count monitoring yielded the lowest ICER, specifically, -145,190 THB (-4,799 USD) per QALYs gained. Thus, the latter was the most cost-effective option based on a ceiling threshold of 160,000 THB (5,289 USD) per QALYs gained, as recommended by the Subcommittee for the Development of the NLEM [10].

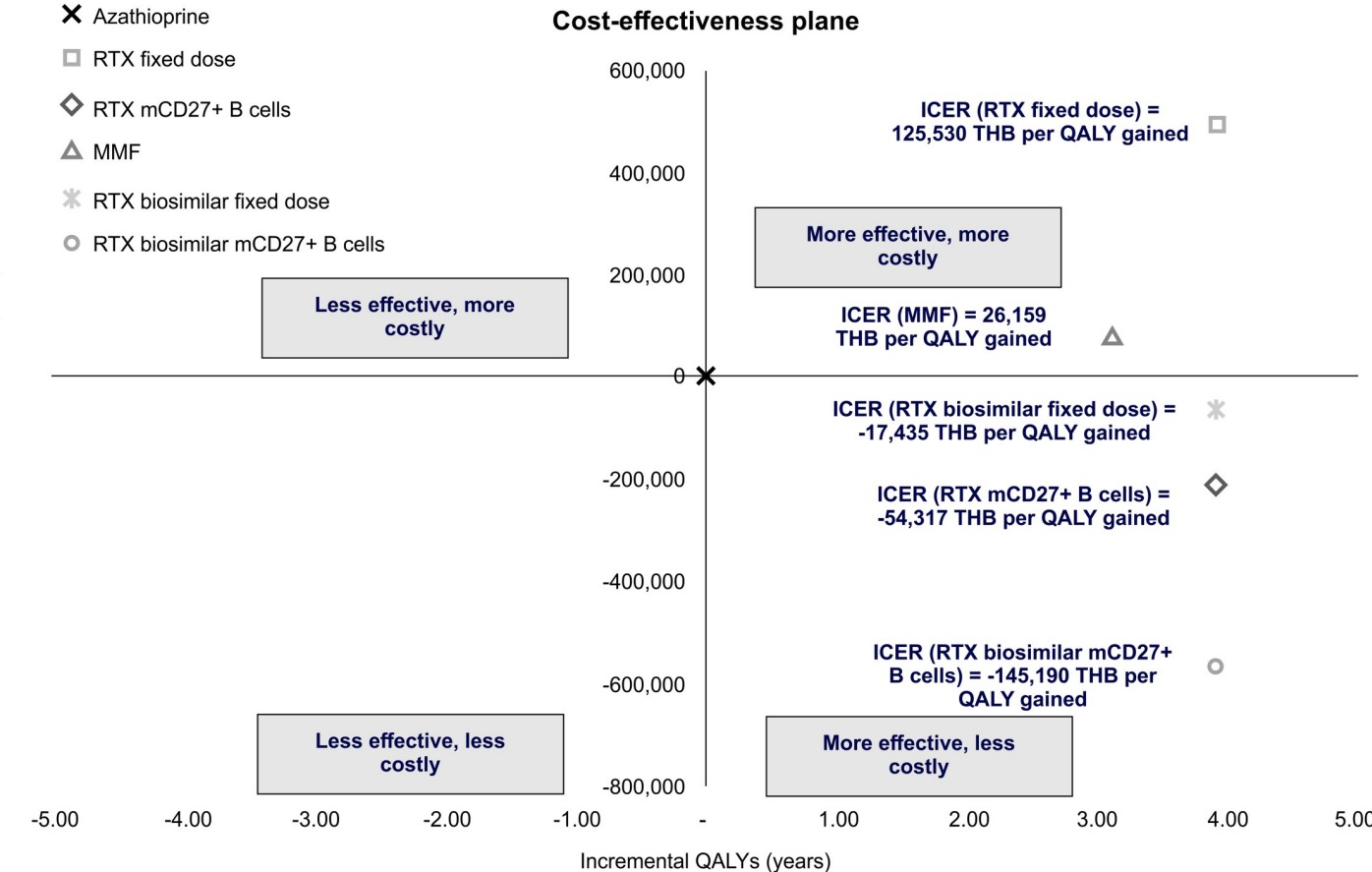

**Fig 2. Cost-effectiveness plane of lifetime cost and effectiveness of six treatment options for NMOSD.** ICER—incremental cost-effectiveness ratio; RTX—rituximab; MMF—mycophenolate mofetil; mCD27+ B cells—CD27+ B cell monitoring. The ICER is demonstrated as the Thai baht per QALY gained. The currency exchange rate for one USD is approximately 30.3 THB.

## Uncertainty analysis

Fig 3 shows the cost-effectiveness acceptability curves for all treatment options. At the willingness-to-pay (WTP) rate of 160,000 THB (5,289 USD) per QALY gained, the results of the PSAs indicate that the administration of the rituximab biosimilar with CD27+ memory B cell monitoring has the highest probability of being cost effective (48%), followed by azathioprine (30%), MMF (13%), and original rituximab with CD27+ memory B cell monitoring (9%). The results of the one-way sensitivity analysis regarding the administration of the rituximab biosimilar with CD27+ memory B cell monitoring are presented in Fig 4. The ICER was most sensitive to variations in the cost due to severe relapse treatment, followed by the efficacy of rituximab in preventing relapse, discount rate for outcome, discount rate for cost, price of biosimilar rituximab, efficacy of rituximab in preventing severe relapse, utility of patients with moderately severe disability, and utility of patients with no or mild disability.

## Sensitivity analysis based on the current price reduction for MMF

Due to the availability of generic MMF in Thailand, the purchasing price from the Prasat Neurological Institute on May 1, 2018, for a 250-mg tablet was 14.5 THB (0.48 USD) compared to 22 THB (0.73 USD) reference price listed in the database of the Drugs and Medical Supplies Information Center [13]. With respect to the MMF evaluation, the 14.5 THB (0.48 USD)

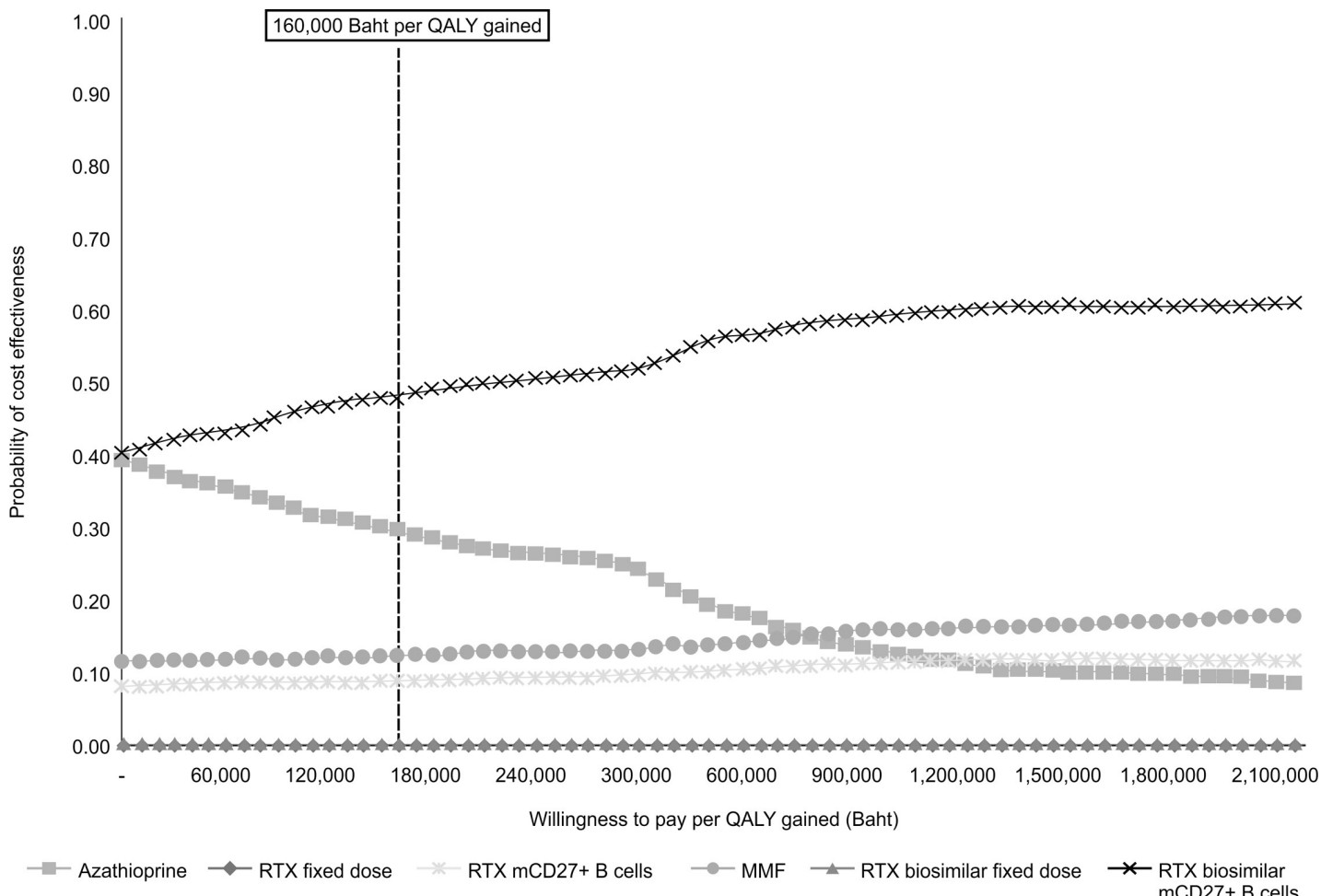

**Fig 3. Acceptability curves of the cost effectiveness at different ceiling thresholds for six NMOSD treatment options.** RTX—rituximab; MMF—mycophenolate mofetil; mCD27+ B cells–monitoring of CD27+ B cells.

tablets were considered in the model. The lifetime cost of MMF for NMOSD treatment decreased from 3,746,932 to 3,151,991 THB (123,866 to 104,198 USD). Accordingly, the new ICER generated by MMF was -164,653 THB (5,443 USD) (S2 Fig), which indicates that the use of MMF at the current price was cost effective, based on a ceiling threshold of 160,000 THB (5,289 USD) per QALY gained. The cost-effectiveness acceptability curves for all treatment options after applying the reduced MMF price are shown in S3 Fig. At the WTP rate of 160,000 THB (5,289 USD) per QALYs gained, the results of the PSA demonstrated that both MMF and azathioprine had a similar probability of being cost effective, i.e., 32%. The rituximab biosimilar and original rituximab administrations with CD27+ memory B cell count monitoring had cost-effectiveness probabilities of 31% and 5%, respectively.

## Budget impact analysis

The budget impact (Table 3) was based on an NMOSD prevalence rate of 0.403 per 100,000 [23], which corresponds to approximately 280 NMOSD patients in a population of 69 million [22]. Azathioprine resistance was found in 30% of the Thai cohorts [24]. Eighty-four patients were identified as resistant to conventional NMOSD drugs. Assuming a coverage treatment

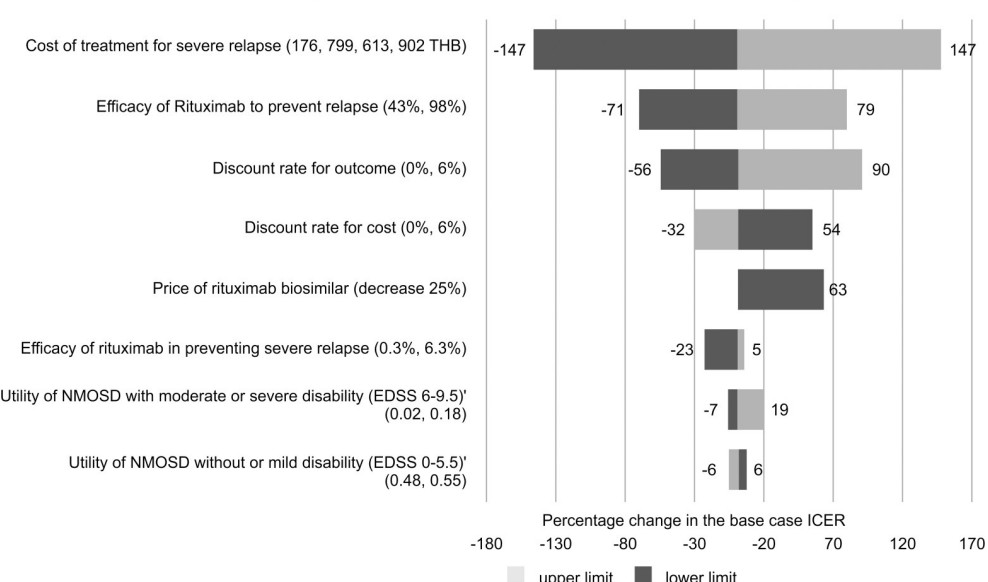

**Fig 4. Results of a one-way sensitivity analysis of the administration of a rituximab biosimilar with a CD27[+] B cells monitoring regimen compared to that of azathioprine in the treatment of NMOSD patients.**

rate of 35% in the first year, 70% in the second year, and 100% in the third year, the approximate incremental budgets for MMF, based on the 14.5 THB price, in the first and second years were 1,123,176 and 2,129,867 THB (37,130 and 70,409 USD), respectively. After the third year, the incremental budget was predicted to stabilize at 2,859,618 THB (94,533 USD). With respect to rituximab biosimilar administration with CD27[+] memory B cell count monitoring, the approximate incremental budgets were 4,511,604, 5,938,445, and 6,604,285 THB (149,144, 196,312, and 218,323 USD) in the first, second, and third years, respectively. The budget was predicted to stabilize after the fourth year at 4,060,047 THB (134,216 USD).

## Discussion

The issues with respect to NMOSD have been acknowledged for more than a decade in Thailand, i.e., since the NMO-IgG test had been made available [25]. However, due to its low disease prevalence, patients are often overlooked and do not receive adequate attention from the

**Table 3. Budget impact of the adoption of new therapies for NMOSD in THB (USD).**

| Treatment | | 1st Year | 2nd Year | 3rd Year | After 3rd Year |
|---|---|---|---|---|---|
| Azathioprine | | 116,485 | 349,456 | 682,272 | 682,272 |
| | | (3,844) | (11,533) | (22,517) | (22,517) |
| MMF | Total price | 1,239,662 | 2,489,323 | 3,541,890 | 3,541,890 |
| | | (40,913) | (82,156) | (116,894) | (116,894) |
| | Incremental budget | 1,123,176 | 2,129,867 | 2,859,618 | 2,859,618 |
| | | (37,069) | (70,293) | (94,377) | (94,377) |
| Biosimilar Rituximab | Total price | 4,628,090 | 6,287,901 | 7,286,557 | 4,742,318 |
| | | (152,742) | (207,521) | (240,480) | (156,152) |
| | Incremental budget | 4,511,604 | 5,938,445 | 6,604,285 | 4,060,047 |
| | | (148,898) | (195,988) | (217,963) | (133,995) |

national health system, e.g., in terms of reimbursement costs for the treatment of severe relapse or high-efficacy drugs for the prevention of relapse. In western countries, rituximab is regarded as the first-line therapy to prevent relapse and is used as a rescue therapy after another first-line therapy has failed. Although there is still no consensus [26,27], patients with NMOSD in western countries often have access to highly efficacious drugs for relapse prevention, with no consideration being given to its cost effectiveness by NMOSD patients. This is a very different situation from low- and middle-income countries where patients who suffer from the failure of first-line medications must continue with the same therapy due to the inaccessibility of high-cost drugs and thus suffer from the consequent inadequate relapse prevention and disability accrual. This is the first study to demonstrate that appropriate regimen adjustments may substantially increase the cost effectiveness of high-efficacy drugs in the treatment of NMOSD. This study provides new evidence regarding the economic impact associated with the extended application of high-efficacy drugs for patients with NMOSD and may be relevant for those making public health policy decisions.

In particular, this study demonstrates that at the current WTP threshold of 160,000 THB (5,289 USD), a rituximab biosimilar or generic MMF at the reduced price of 14.5 THB (0.48 USD)/250-mg tablet may be as cost effective as the current practice if the dose is calibrated based on disease activity as assessed by the $CD27^+$ memory B cell count in the context of the Thai health care system. Our analysis further indicates that appropriate regimens of high-efficacy drugs may avoid the extra costs required for relapse treatment associated with plasmapheresis and prolonged hospitalization. We compared our study with a previous report focusing on the economic analysis of another idiopathic CNS demyelination in Thailand, i.e., multiple sclerosis (MS) [28]. The latter study demonstrated that the disease modified therapy for MS was not cost effective. However, in our study, the low cost of the rituximab biosimilar (one-tenth the current price of the disease modified therapy for MS) resulted in both cost effectiveness and cost savings. Similarly, MMF proved cost effective in our model, as the price of generic MMF is as low as 35% that of the standard price. Thus, for high-efficacy drugs, price is an important factor for cost effectiveness, especially in low- and middle-income countries.

A strength of our study is the classification of NMOSD relapse severity, which correlates with the actual cost of treatment, especially for patients who experience relapses during therapy. Patients with severe relapses require high-cost treatments, including plasmapheresis, and some may also require ventilation support because of respiratory failure. Treatment costs may also include those caused by complications related to prolonged hospitalizations, such as ventilator-associated pneumonia, urinary tract infections, and deep venous thrombosis. All of these additional costs could be avoided by preventing relapses or decreasing relapse rates in NMOSD patients.

In our model, the one-way sensitivity analysis demonstrated that the management of severe relapse had a strong impact on the ICER, thus reflecting the relatively high costs for this treatment. The wide range of 95% CI in our data, i.e., 176,799–613,902 THB (5,844–20,924 USD) reflects the individual differences among patients with severe relapse, such as the length of hospitalization or various complications. Another factor influencing the ICER is the efficacy of rituximab in relapse prevention, as patients with severe and mild relapse exhibited decreased utility, i.e., 0.29 and 0.07, respectively. Severe relapse also causes disease progression and, in some cases, death during the attack, all of which reflect the natural history and progression of NMOSD [29] and the high costs associated with severe relapse treatment. These data confirm that relapse prevention is the mainstay of treatment for patients with NMOSD.

Our budget impact analysis was based on the assumption that treatment coverage was 35% per year for patients who relapsed under conventional therapy. Nonetheless, there might be some variations in the budget impacts during the first three years to cover 35% to 100% of the

patients with NMOSD, thus resulting in a 2- to 3-fold budget increase compared to the estimate, after which the budget impact is predicted to stabilize at approximately 3–4 million THB per year.

There are some limitations to this study. First, our model did not account for costs related to the adverse effects of medications. Second, the cost of rituximab treatment was calculated by assuming two additional administrations per year, whereas some patients may require more frequent administrations during the first few years. However, data from a 5-year follow-up study revealed that, on average, rituximab is administered eight times in five years (1.6 times per year) [11], which indicates that the actual drug cost may be even lower than that hypothesized by our model. Recent published data classified patients with NMOSD as clustered and nonclustered relapse patients, according to the number of relapse occurrences within 12 months of their previous attack [30]. In patients with cluster relapse, the high efficacy of medication should be considered because of the frequent relapses, whereas in the nonclustered group, the relapses were infrequent. These data indicate that it is possible to decrease preventive medication to lower costs during the nonclustered period. If we considered this strategy in our model, the cost of medication would be lower, and there would be a cost savings in the cost-effective analysis. Third, since NMOSD is a rare disease, some of the parameters may not be found in the literature, including the probability of death due to relapse [16], the probability of severe relapse requiring plasmapheresis [16], and the probability of disease progression after severe relapse [4]. These parameters were analyzed at our institute, which is a tertiary referral neurological center, so there is selection bias, particularly when considering the effect of the high proportion of patients with severe relapse. Fourth, we did not consider changes in therapy due to treatment failure, even though such changes may occur in clinical practice [6–9,31]. Fifth, including the different cohorts in the cost calculations, i.e., cohorts where 13% of the patients are in seronegative NMOSD groups and prospective cohorts are in transition, the probability calculations that include all seropositive NMOSD will influence the results. In other words, cost parameters generated by patient groups of mixed populations of seronegative NMOSD who exhibited lower relapse rates will lower the cost parameters more than expected. The patients in the seropositive NMOSD group who more frequently relapse will incur higher costs for both direct and indirect medical expenses compared with the seronegative patients. However, if we apply only AQP4 positive NMOSD in the cost calculations, using high efficacy medication will result in greater differences in reducing relapse frequency and in further reducing the costs associated with treatments. Hence, the ICER that generates this situation negatively impacts our results, which indicates greater cost savings when treating NMOSD with high efficacy medications. Finally, since disease relapse may occur 10 years after remission, a lifetime treatment period was applied in our model [32]. However, some of the patients might cease immunosuppressive treatment after disease onset, especially patients with prolonged disease quiescence [33]. An additional limitation in the budget impact estimation was that the hospital-based prevalence data might not reflect the prevalence of NMOSD patients in the general population. Furthermore, our estimation did not include newly diagnosed NMOSD cases because they exhibit a very low incidence rate in the patient population. Notably, only NMOSD patients who were drug-refractory for at least six months following their initial diagnosis were defined as drug resistant.

In conclusion, this study demonstrated that, in the context of the Thailand healthcare system, treatment with a rituximab biosimilar combined with disease activity monitoring of the CD27$^+$ memory B cell count or treatment with a generic MMF were cost efficient and exhibited a high probability of being cost effective when compared with the current practice. The estimated budget impact of treating patients with NMOSD who are resistant to conventional therapy is 1–6 million THB (33,000–198,000 USD) during the first three years, after which the

budget stabilizes at 3 to 4 million THB (99,000–132,000 USD), for MMF- and rituximab-based treatments, respectively. This study may encourage politicians to extend the indications of high-efficacy drugs for the prevention of NMOSD relapse to include rituximab and mycophenolate mofetil in the NLEM.

## Supporting information

**S1 Fig. Meta-analysis of mycophenolate mofetil in comparison to azathioprine in relapse-free NMOSD.**
(TIF)

**S2 Fig. Cost-effectiveness plane.** Cost-effectiveness plane covering lifetime cost effectiveness of six treatment options for NMOSD, after MMF price adjustment to 14.5 THB. RTX: rituximab; MMF: mycophenolate mofetil; mCD27$^+$ B cells: monitor CD27$^+$ B cells.
(TIF)

**S3 Fig. Cost-effectiveness acceptability curves.** Acceptability curves of cost effectiveness at the different ceiling thresholds for six NMOSD treatment options, after the MMF price adjustment to 14.5 THB. RTX: rituximab; MMF: mycophenolate mofetil; mCD27$^+$ B cells: monitor CD27$^+$ B cells.
(TIF)

**S1 Text. Treatment efficacy of MMF.**
(DOCX)

## Acknowledgments

This project is a continuation of the research from the 15$^{th}$ Health Economic Evaluation Training Course organized by the Health Intervention and Technology Assessment Program (HITAP). The authors would like to thank Dr. Pritaporn Kingkaew (HITAP), who served as a consultant on this project.

## Author Contributions

**Conceptualization:** Saharat Aungsumart, Metha Apiwattanakul.

**Data curation:** Saharat Aungsumart.

**Formal analysis:** Saharat Aungsumart.

**Investigation:** Saharat Aungsumart.

**Methodology:** Saharat Aungsumart.

**Project administration:** Saharat Aungsumart.

**Supervision:** Metha Apiwattanakul.

**Writing – original draft:** Saharat Aungsumart.

**Writing – review & editing:** Saharat Aungsumart, Metha Apiwattanakul.

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
