## [Decision Letter · Decision Letter 0]

10 Dec 2019

PONE-D-19-28259

Cost-effectiveness of rituximab and mycophenolate mofetil for neuromyelitis optica spectrum disorder in Thailand:Economic evaluation and budget impact analysis

PLOS ONE

Dear Dr. Aungsumart,

Thank you for submitting your manuscript to PLOS ONE. After careful consideration, we feel that it has merit but does not fully meet PLOS ONE’s publication criteria as it currently stands. Therefore, we invite you to submit a revised version of the manuscript that addresses the points raised during the review process.

We would appreciate receiving your revised manuscript by Jan 24 2020 11:59PM. To enhance the reproducibility of your results, we recommend that if applicable you deposit your laboratory protocols in protocols.io, where a protocol can be assigned its own identifier (DOI) such that it can be cited independently in the future. For instructions see: http://journals.plos.org/plosone/s/submission-guidelines#loc-laboratory-protocols

We look forward to receiving your revised manuscript.

Kind regards,

Massimo Filippi

Academic Editor

PLOS ONE

Journal Requirements:

2. Your ethics statement must appear in the Methods section of your manuscript. If your ethics statement is written in any section besides the Methods, please move it to the Methods section and delete it from any other section. Please also ensure that your ethics statement is included in your manuscript, as the ethics section of your online submission will not be published alongside your manuscript.

Reviewers' comments:

Reviewer's Responses to Questions

**Comments to the Author**

1. Is the manuscript technically sound, and do the data support the conclusions?

Reviewer #1: Yes

Reviewer #2: Partly

2. Has the statistical analysis been performed appropriately and rigorously? 

Reviewer #1: Yes

Reviewer #2: I Don't Know

3. Have the authors made all data underlying the findings in their manuscript fully available?

Reviewer #1: Yes

Reviewer #2: No

4. Is the manuscript presented in an intelligible fashion and written in standard English?

Reviewer #1: No

Reviewer #2: No

5. Review Comments to the Author

Reviewer #1: In this work Angusmart and Apiwattanakul perform a cost-effectiveness analysis of different regimens of mycophenolate mofetil (MMF) and rituximab in neuromyelitis optica spectrum disorders patients resistant to azathioprine in Thailand. Authors conclude that rituximab has the highest quality-adjusted life year (QALY) and the better cost-effectiveness profile when its biosimilar is administered according to B cell CD27+ monitoring. Moreover, also generic MMF is cost-saving and effective. In this perspective, Authors suggest the inclusion of both rituximab biosimilar and generic MMF in their national drug list of essential medicines.

I agree with Authors and with their analysis. In NMOSD disability accrual is usually associated with relapses, so, it is likely that an adequate preventive therapy would reduce disability, hospitalization and their related costs. I have only several points which need a further clarification.

These are my comments.

Major points:

1. I have several concerns regarding the rituximab administration “1 g twice a week for six months”. Could you please add a reference? It is completely different from the usual regimen (1 g every six months). This latter approach would even ameliorate your cost-effectiveness analysis.

2. Authors should add a paragraph inclusive of a detailed description of NMOSD patients from the Prasat Neurological Institute who were used for the analysis, specifying the type of analysis (i.e. cost variables evaluation, clinical variable evaluation). For instance, it is not clear why Authors used 36 patients for the evaluation of direct medical and non-medical costs and 49 for the estimation of transition probabilities of relapses and death). Moreover, it is never specified whether all patients are seropositive and their annual relapse rate.

3. Add a definition of relapse severity (i.e. a severe relapse requires both steroid treatment and plasma exchange and an EDSS of XX). I suppose you used such definitions to infer direct medical and nonmedical costs from the Prasat Neurological Institute?

4. Figures quality: Authors should provide better figures of higher quality (they appear grainy), should not use symbols like “_” between words and a color scale would make the meaning of pictures more immediate. Moreover, pay attention to list the entire acronyms in figures legends.

Minor points:

1. Introduction: add a reference when state that severe relapse is treated with high-dose steroids followed by plasma exchange (according to Abboud et al., MSJ 2016, plasma exchange should be started early).

2. Introduction: MMF and NLEM were never cited “in extensor” before the acronym

3. Introduction: “Not only for the number of relapse but also the relapse severity”. Consider reword the sentence (it has no verbal predicate).

4. In order to favour the comprehension of the work also among Europeans and Americans, I would add the conversion of Thai Bath in Western currency (i.e. euros or dollars).

5. Cost variables: “The indirect costs not include in the study to avoid double counting”. Please reword the sentence (what do you mean? That indirect costs was not included?).

6. Table 3: NMOSDD should be changed in NMOSD.

7. Discussion: “Although the burden of NMOSD was known and recognized disease more han decate in Thailand.” Consider to write a new sentence, this lacks predicates.

8. Discussion: I would change “policy makers” into politicians.

Reviewer #2: General comment:

At first glance, the manuscript seemed to deal with a very regional topic and not suitable for the journal. However, after consideration, I feel the manuscript is dealing with an important topic in the field of medical economics with potential for further development. Although the model they built contains many flaws from clinical viewpoint, this type of research may expand the possibility of future researches in the field of clinical neurology. Nevertheless, whether this study is “scientific” or not is unsure.

Major comments:

1. The English throughout the manuscript is not well written. There are many grammatical errors. I strongly recommend the authors to undergo a native check.

2. Was there no previous report or systematic review that evaluated the effect of long-term low-dose oral prednisolone? I heard that oral PSL is in the first-line in some countries. If possible, I want to know the location of oral PSL on the plane of Figure 2.

3. What kind of statistical software did you utilize for this study? Matlab or SPSS? Please clarify in the manuscript.

4. What is the mainstay of relapse prevention for NMOSD patients in Western countries? Do they also utilize azathioprine for the first line therapy? Please describe in the introduction or in the discussion.

5. Where are the legends for figures and tables? Please spell out the abbreviations in the figures and tables.

Minor comments:

1. Page 4: “rituximab (1gm)” is to be changed to “(1mg)”?

2. Page 4-5: Is there an evidence to divide “mild” and “moderate” disability at EDSS 5.5? EDSS 5.5 seems more than “moderate” disability to me. I am not sure about this point, so you can ignore this comment if it is too hard to answer.

3. Page 5: Is it OK not to divide the relapse phenotypes into ON (optic neuritis), myelitis, and CVOs lesions? EDSS scoring deals with this point on the same dimension, but this could cause some bias to your research.

4. Page 6: Were all enrolled NMOSD patients positive for the serum anti-AQP4 antibody?

Were they also screened for serum anti-MOG antibody?

5. Page 6: As to the cost, did you take the cost of regularly monitoring CD27+ memory B cells? Is it easily monitored in all tertiary hospitals without paying extra fee in your country?

6. Page 7: The size of 49 looks too small to assume the transition probability. Can’t you increase the size by expanding the enrollment period? (Although it is usually not allowed in scientific researches).

7. It is presented that one patient died due to severe attack. To me, it seems that death for attack in NMOSD patients became quite rare these days. Patients with attacks treated in advanced hospitals such as yours seem not to die so frequently. Did the patient really die from an attack? Respiratory dysfunction based on myelitis?

8. Page 9: Are the values of “25.49”, “24.29”, and “25.34” significantly different? What are the 95% CI of these values, if any? These values look only a trivial difference to me.

9. Page 10: Table 2. What is “Increment al cost”? “Al cost” does not look like an English word.

10. Page 11: “azathioprine at (30%)” does not need “at”.

11. Page 12: Title of Table 3. What is “NMOSDD”?

12. Page 14: “which reflects the natural history of NMOSD (27)”. I think the manuscript is published more about 20 years before, which is far before the discovery of serum anti-AQP4 antibody. Although you need not to remove the reference, is the “die during attacks” really a natural history in NMOSD?

13. Page 15: A recently published manuscript reported that NMOSD patients may show uneven distribution of attack occurrence (PMID: 31757816). Does this new insight possibly affect your model or future research perspective? Or such attack occurrence unevenness does not affect your conclusions? I want to read a discussion about the possible effect of such attack unevenness from your medical-economic viewpoint, if possible.

14. Page 16. “Furthermore, our estimation … very low incidence in the patient population” Although it is true that initial attack account for only a small proportion, initial attack occurrence could affect the clinical course or prognosis in NMOSD. Usually, the initial treatment is likely to be continued for the moment unless the patients experience repeated severe attacks. Thus, therapeutic strategy for newly diagnosed NMOSD patients seems to be also important. I recommend the authors to include the newly diagnosed NMOSD patients in their future researches.

15. Figure 1. This figure that shows the used Markov model should be drawn more decently. It looks like a mere sketch illustration made by PowerPoint.

16. Figure 2. Most readers may be unable to interpret this figure. What do the X-axis and Y-axis clinically imply? To which axis or dimension does the description of "More effective, less costly" belong? Lower-right side?

6. PLOS authors have the option to publish the peer review history of their article (what does this mean?). If published, this will include your full peer review and any attached files.

Reviewer #1: No

Reviewer #2: No

---

## [Author Response · Author response to Decision Letter 0]

2 Jan 2020

Reviewer1

Major points:

1. I have several concerns regarding the rituximab administration “1 g twice a week for six months”. Could you please add a reference? It is completely different from the usual regimen (1 g every six months). This latter approach would even ameliorate your cost-effectiveness analysis.

 The actual administration of rituximab in our model is “induction via intravenous rituximab at 1000 mg two weeks apart followed by 1000 mg every six months.” We have fixed this mistake in page 4 of the manuscript.

2. Authors should add a paragraph inclusive of a detailed description of NMOSD patients from the Prasat Neurological Institute who were used for the analysis, specifying the type of analysis (i.e. cost variables evaluation, clinical variable evaluation). For instance, it is not clear why Authors used 36 patients for the evaluation of direct medical and non-medical costs and 49 for the estimation of transition probabilities of relapses and death). Moreover, it is never specified whether all patients are seropositive and their annual relapse rate.

 The cohort of 36 patients are included in the cost variable evaluation because these data already existed retrospectively. We used another cohort of 49 patients, instead of the 36 patients in the previous study in order to determine the transition probability because data from the 49 patients were collected in prospective manner which will be more realistic for the incidence of relapse, proportion of severe relapse, and death due to NMOSD. We added these data and included the demographic information of the second cohort in the manuscript. We included a statement to this effect on page 6 and 8 of the manuscript.

3. Add a definition of relapse severity (i.e. a severe relapse requires both steroid treatment and plasma exchange and an EDSS of XX). I suppose you used such definitions to infer direct medical and nonmedical costs from the Prasat Neurological Institute?

 The definition of severe relapse was added in the manuscript on page 5, “Patients with severe relapse, which is defined by severe disability that was sustained or worsened after high dose steroids as indicated by Expanded Disability Status Scale (EDSS) scores ≥7.0 in patients who presented with myelitis or a visual acuity worse than 20/200 in patients who presented with optic neuritis”. We used the same definition in this study for direct medical and nonmedical costs. We have included this statement on page 5 of the paper

4. Figures quality: Authors should provide better figures of higher quality (they appear grainy), should not use symbols like “_” between words and a color scale would make the meaning of pictures more immediate. Moreover, pay attention to list the entire acronyms in figures legends.

 We have replaced the figures and added acronyms in figure legends. 

Minor points:

1. Introduction: add a reference when state that severe relapse is treated with high-dose steroids followed by plasma exchange (according to Abboud et al., MSJ 2016, plasma exchange should be started early).

 This has been done. 

2. Introduction: MMF and NLEM were never cited “in extensor” before the acronym

 We have fixed this error. 

3. Introduction: “Not only for the number of relapse but also the relapse severity”. Consider reword the sentence (it has no verbal predicate).

 This has been done.

4. In order to favour the comprehension of the work also among Europeans and Americans, I would add the conversion of Thai Bath in Western currency (i.e. euros or dollars).

 USD currency values were added in the manuscript.

5. Cost variables: “The indirect costs not include in the study to avoid double counting”. Please reword the sentence (what do you mean? That indirect costs was not included?).

 This study did not include the indirect cost for analysis, following the HTA guideline of Thailand. The Indirect cost, such as the opportunity costs of patient and caregivers are not included in the study because these data have already been considered in the utility and QALY of patients. This data was referenced according to the “Avoiding Double-Counting in Pharmacoeconomic Studies” article in Pharmacoeconomics 1997 May; 11 (5): 385-388.

6. Table 3: NMOSDD should be changed in NMOSD.

 This was a typing error. We have fixed this mistake.

7. Discussion: “Although the burden of NMOSD was known and recognized disease more than decate in Thailand.” Consider to write a new sentence, this lacks predicates.

 We changed this to, “The burden of NMOSD was recognized more than decade ago in Thailand since NMO-IgG test was available”

8. Discussion: I would change “policy makers” into politicians.

 We have done this. 

Reviewer2

Major comments

1. The English throughout the manuscript is not well written. There are many grammatical errors. I strongly recommend the authors to undergo a native check.

This manuscript was sent for professional editing by Editage after we completed revisions.

2. Was there no previous report or systematic review that evaluated the effect of long-term low-dose oral prednisolone? I heard that oral PSL is in the first-line in some countries. If possible, I want to know the location of oral PSL on the plane of Figure 2.

You are correct. There is a study that showed steroids are effective in preventing NMOSD relapse (PMID 17623727). However, the sample size in the study was small (n = 11), and there was no RCT or systematic review. It is generally accepted that azathioprine is more effective and has less side effects. In our country, patients are typically administered azathioprine with or without steroids for relapse prevention, therefore we do not have data of PSL only.

3. What kind of statistical software did you utilize for this study? Matlab or SPSS? Please clarify in the manuscript.

All data were collected and analyzed using Microsoft Excel 2016 (License number QY49T 8WCQ7 433XX HPBCM 6DQDV). We have added this statement to the Material and Methods section, page 4.

4. What is the mainstay of relapse prevention for NMOSD patients in Western countries? Do they also utilize azathioprine for the first line therapy? Please describe in the introduction or in the discussion.

 In Western countries, rituximab is considered as the first line therapy to prevent relapse or for use as a rescue therapy after failing another first line therapy. Although there is still no consensus, patients with NMOSD in that region often have to access highly efficacious drugs for relapse prevention, without considering the cost effectiveness in NMOSD patients. This is a very different situation from low-middle income countries. Patients who suffer from failure of first line medication still have to continue with the same therapy due to inaccessible to high cost drugs. Unfortunately, patients are often ultimately disabled given the inadequate relapse prevention. 

 This data was added to the Discussion section of the manuscript on page 14.

5. Where are the legends for figures and tables? Please spell out the abbreviations in the figures and tables.

 The figure legends were inserted after figure titles in the manuscript.

Minor comments:

1. Page 4: “rituximab (1gm)” is to be changed to “(1mg)”?

 The actual dose of rituximab is 1gm. We adjusted this to 1000 mg for clarity.

2. Page 4-5: Is there an evidence to divide “mild” and “moderate” disability at EDSS 5.5? EDSS 5.5 seems more than “moderate” disability to me. I am not sure about this point, so you can ignore this comment if it is too hard to answer.

 We use EDSS 5.5 because an EDSS greater than 6.0 is considered a severe disability.

3. Page 5: Is it OK not to divide the relapse phenotypes into ON (optic neuritis), myelitis, and CVOs lesions? EDSS scoring deals with this point on the same dimension, but this could cause some bias to your research.

 We agree with you. To use several clinical relapse phenotype would increase the accuracy of our model. However the visual acuity measurement is also included in the EDSS functional score even that it is not sensitive enough to show the difference after converting to EDSS. And the majority of our patients were myelitis that can be captured by EDSS change. The EDSS is also the standard measurement in most publication, so we decided to use EDSS as the measurement for all clinical attack phenotypes. None of our patients had CVO symptoms alone, they also had either myelitis or optic neuritis as the major clinical relapses which need more aggressive therapy apart from steroid alone.

4. Page 6: Were all enrolled NMOSD patients positive for the serum anti-AQP4 antibody?

Were they also screened for serum anti-MOG antibody?

 Not all patients were positive for AQP4 antibody. In total, 87% of patients in the plasma exchange cohort were AQP4 seropositive (36 patients). The prospective cohort was 100% AQP4 seropositive. We added this information to the manuscript. There was no data on MOG antibody status in either cohort because the test was not available at that time. 

5. Page 6: As to the cost, did you take the cost of regularly monitoring CD27+ memory B cells? Is it easily monitored in all tertiary hospitals without paying extra fee in your country?

 Yes, we accounted for the cost of regularly monitoring CD27+ memory B cells in the model.

6. Page 7: The size of 49 looks too small to assume the transition probability. Can’t you increase the size by expanding the enrollment period? (Although it is usually not allowed in scientific researches).

 Yes, we agree that 49 patients may not provide high power to assume the transition probability. However these are the data that we can collect prospectively during that period of time.

7. It is presented that one patient died due to severe attack. To me, it seems that death for attack in NMOSD patients became quite rare these days. Patients with attacks treated in advanced hospitals such as yours seem not to die so frequently. Did the patient really die from an attack? Respiratory dysfunction based on myelitis?

 This patients did not die after an acute attack. After the high cervical cord lesion as a consequence of severe relapse, she became ventilator dependent and developed ventilator-associated pneumonia and death due to sepsis. So we consider this death due to severe relapse. 

8. Page 9: Are the values of “25.49”, “24.29”, and “25.34” significantly different? What are the 95% CI of these values, if any? These values look only a trivial difference to me.

 These values in the table are life years (LYs) and were not used to compare each other. The LYs were used to calculate ICERs, which are the final outcome of our study. Since Lys and ICER were not used to compare for the statistically significant difference, so the 95% CI of these values cannot be calculated.

9. Page 10: Table 2. What is “Increment al cost”? “Al cost” does not look like an English word.

 This is typing error. We have corrected it. 

10. Page 11: “azathioprine at (30%)” does not need “at”.

 This is typing error. We have corrected it.

11. Page 12: Title of Table 3. What is “NMOSDD”?

 This is typing error. We have corrected it. 

12. Page 14: “which reflects the natural history of NMOSD (27)”. I think the manuscript is published more about 20 years before, which is far before the discovery of serum anti-AQP4 antibody. Although you need not to remove the reference, is the “die during attacks” really a natural history in NMOSD?

 We agree that death due to a NMOSD relapse is rare given the current treatment. However, some patients who have ongoing relapse due to “unsuitable relapse prevention” is not uncommon. Severe relapse leads to an accumulation of ailments, which may cause complications, such as pressure sores, UTI, pneumonia, and sepsis leading to death or disability.

13. Page 15: A recently published manuscript reported that NMOSD patients may show uneven distribution of attack occurrence (PMID: 31757816). Does this new insight possibly affect your model or future research perspective? Or such attack occurrence unevenness does not affect your conclusions? I want to read a discussion about the possible effect of such attack unevenness from your medical-economic viewpoint, if possible.

 We added the following information to the manuscript on page 16-17. “Recent published data classified patients with NMOSD as clustered and nonclustered relapse, according to occurrences of relapse within 12 months from their previous attack (30). In patients with clustered relapse, the high efficacy of medication should be considered because of the frequent relapses, whereas in the nonclustered group, the relapse was infrequent. This data shows that it is possible to decrease preventive medication in order to lower costs during the nonclustered period. If we considered this strategy in our model, the cost of medication would be lower and there would be a cost savings in the cost effective analysis”.

14. Page 16. “Furthermore, our estimation … very low incidence in the patient population” Although it is true that initial attack account for only a small proportion, initial attack occurrence could affect the clinical course or prognosis in NMOSD. Usually, the initial treatment is likely to be continued for the moment unless the patients experience repeated severe attacks. Thus, therapeutic strategy for newly diagnosed NMOSD patients seems to be also important. I recommend the authors to include the newly diagnosed NMOSD patients in their future researches.

 We agree with you and will include the newly diagnosed NMOSD patient in future research.

15. Figure 1. This figure that shows the used Markov model should be drawn more decently. It looks like a mere sketch illustration made by PowerPoint.

 This has been fixed. 

16. Figure 2. Most readers may be unable to interpret this figure. What do the X-axis and Y-axis clinically imply? To which axis or dimension does the description of "More effective, less costly" belong? Lower-right side?

 Figure 2 is a cost effective plane study. The X-axis is Incremental QALYs (years) and Y-axis is Incremental cost (THB). We added a dialogue box of effectiveness and cost in all of the 4 Quadrant for clarity.

---

## [Decision Letter · Decision Letter 1]

8 Jan 2020

PONE-D-19-28259R1

Cost-effectiveness of rituximab and mycophenolate mofetil for neuromyelitis optica spectrum disorder in Thailand:Economic evaluation and budget impact analysis

PLOS ONE

Dear Dr. Aungsumart,

Thank you for submitting your manuscript to PLOS ONE. After careful consideration, we feel that it has merit but does not fully meet PLOS ONE’s publication criteria as it currently stands. Therefore, we invite you to submit a revised version of the manuscript that addresses the points raised during the review process.

We would appreciate receiving your revised manuscript by Feb 22 2020 11:59PM. To enhance the reproducibility of your results, we recommend that if applicable you deposit your laboratory protocols in protocols.io, where a protocol can be assigned its own identifier (DOI) such that it can be cited independently in the future. For instructions see: http://journals.plos.org/plosone/s/submission-guidelines#loc-laboratory-protocols

We look forward to receiving your revised manuscript.

Kind regards,

Massimo Filippi

Academic Editor

PLOS ONE

Reviewers' comments:

Reviewer's Responses to Questions

**Comments to the Author**

1. If the authors have adequately addressed your comments raised in a previous round of review and you feel that this manuscript is now acceptable for publication, you may indicate that here to bypass the “Comments to the Author” section, enter your conflict of interest statement in the “Confidential to Editor” section, and submit your "Accept" recommendation.

Reviewer #1: (No Response)

Reviewer #2: All comments have been addressed

2. Is the manuscript technically sound, and do the data support the conclusions?

Reviewer #1: Yes

Reviewer #2: Yes

3. Has the statistical analysis been performed appropriately and rigorously? 

Reviewer #1: Yes

Reviewer #2: I Don't Know

4. Have the authors made all data underlying the findings in their manuscript fully available?

Reviewer #1: Yes

Reviewer #2: Yes

5. Is the manuscript presented in an intelligible fashion and written in standard English?

Reviewer #1: No

Reviewer #2: Yes

6. Review Comments to the Author

Reviewer #1: Authors adequately replied to the points of the previous revision.

I have only few considerations (see below). Moreover, I highly recommend an accurate revision of English (I doubt a professional editing was performed, as, basically, no language corrections are present in the revised manuscript).

Major points:

1. Cost variables: Unfortunately, as 13% of patients are seronegative (and this population could have a different relapse-rate), Authors should at least add data regarding seronegative patients (do the ARR and disease severity differ from seropositive patients?). As the longitudinal population includes seropositive patients only, this should be taken into account in the statistical analysis or this point should at least be discussed.

Minor points:

1. THB and USD are not spelled out in the Abstract section.

2. Introduction: “NMOSD is a devastating CNS inflammatory demyelination”. Please, prefer “NMOSD is a devastating CNS inflammatory demyelinating disease”.

3. Material and methods: “Which began induction via intravenous rituximab [...]”. Please, revise the syntax of this sentence.

4. Material and methods: USD is not spelled out. Moreover, I would move the exchange rate sentence to the beginning of the “Cost variables” section.

5. Clinical variables: I thank the Authors for adding the clinical and demographic features of the longitudinal NMOSD cohort, but please, revise the English.

6. Please, add USD to Table 2, Table 3 and page 16.

7. “The burden of NMOSD was recognized more than decade in Thailand”. Please, revise the English.

8. Discussion: “[…] inaccessible to high cost drugs. Unfortunately […]”. Please, consider revising (i.e., “[…] inaccessible to high cost drugs with consequent inadequate relapse prevention and disability accrual.”)

9. Discussion: “[…] which helped represent […]”. Consider revising (i.e. “[…] which helped representing […]”).

10. Please correct “these parameters was analysed” in “were analysed”.

Reviewer #2: I think that the authors appropriately responded to the comments and concerns raised from my part. I feel that the revised manuscript has been much improved from the original one.

7. PLOS authors have the option to publish the peer review history of their article (what does this mean?). If published, this will include your full peer review and any attached files.

Reviewer #1: No

Reviewer #2: No

---

## [Author Response · Author response to Decision Letter 1]

22 Jan 2020

Reviewer #1: Authors adequately replied to the points of the previous revision.

I have only few considerations (see below). Moreover, I highly recommend an accurate revision of English (I doubt a professional editing was performed, as, basically, no language corrections are present in the revised manuscript).

Major points:

1. Cost variables: Unfortunately, as 13% of patients are seronegative (and this population could have a different relapse-rate), Authors should at least add data regarding seronegative patients (do the ARR and disease severity differ from seropositive patients?). As the longitudinal population includes seropositive patients only, this should be taken into account in the statistical analysis or this point should at least be discussed.

Answer: We have added the demographic data of the seronegative patients to page 7 of the manuscript. Moreover, this point was added to the discussion section on page 18: “In other words, cost parameters generated by patient groups of mixed populations of seronegative NMOSD who exhibited lower relapse rates will lower the cost parameters more than expected. Those patients in the seropositive NMOSD group who more frequently relapse will incur higher costs for both direct and indirect medical expenses compared with the seronegative patients. However, if we apply only AQP4 positive NMOSD in the cost calculations, using high efficacy medication will result in greater differences in reducing relapse frequency and in further reducing the costs associated with treatments. Hence, the ICER that generates this situation negatively impacts our results, which indicates greater cost savings when treating NMOSD with high efficacy medications”.

Minor points:

1. THB and USD are not spelled out in the Abstract section.

 This has been done.

2. Introduction: “NMOSD is a devastating CNS inflammatory demyelination”. Please, prefer “NMOSD is a devastating CNS inflammatory demyelinating disease”.

 This has been done.

3. Material and methods: “Which began induction via intravenous rituximab [...]”. Please, revise the syntax of this sentence. 

 This sentence was revised to read, “which began with the introduction of two 1000 mg intravenous rituximab doses two weeks apart followed by 1000 mg intravenous rituximab doses every six months.”

4. Material and methods: USD is not spelled out. Moreover, I would move the exchange rate sentence to the beginning of the “Cost variables” section.

 This has been done.

5. Clinical variables: I thank the Authors for adding the clinical and demographic features of the longitudinal NMOSD cohort, but please, revise the English.

 This manuscript was sent to a new company (Elsevier language editing company) for English language editing after we completed the revisions. The certificate is attached to this revision.

6. Please, add USD to Table 2, Table 3 and page 16.

 This has been done.

7. “The burden of NMOSD was recognized more than decade in Thailand”. Please, revise the English.

This sentence was revised as follows: “The issues with respect to NMOSD have been acknowledged for more than a decade in Thailand, i.e., since the NMO-IgG test had been made available…”

8. Discussion: “[…] inaccessible to high cost drugs. Unfortunately […]”. Please, consider revising (i.e., “[…] inaccessible to high cost drugs with consequent inadequate relapse prevention and disability accrual.”)

 This has been done.

9. Discussion: “[…] which helped represent […]”. Consider revising (i.e. “[…] which helped representing […]”).

 This has been done.

10. Please correct “these parameters was analysed” in “were analysed”.

 This has been done.

---

## [Editor Report · Decision Letter 2]

29 Jan 2020

Cost effectiveness of rituximab and mycophenolate mofetil for neuromyelitis optica spectrum disorder in Thailand Economic evaluation and budget impact analysis

PONE-D-19-28259R2

Dear Dr. Aungsumart,

We are pleased to inform you that your manuscript has been judged scientifically suitable for publication and will be formally accepted for publication once it complies with all outstanding technical requirements.

With kind regards,

Massimo Filippi

Academic Editor

PLOS ONE
---

## [Editor Report · Acceptance letter]

5 Feb 2020

PONE-D-19-28259R2 

Cost effectiveness of rituximab and mycophenolate mofetil for neuromyelitis optica spectrum disorder in Thailand: Economic evaluation and budget impact analysis 

Dear Dr. Aungsumart:

I am pleased to inform you that your manuscript has been deemed suitable for publication in PLOS ONE. Congratulations! Your manuscript is now with our production department. 

With kind regards,

on behalf of

Prof. Massimo Filippi 

Academic Editor

PLOS ONE